# Early Leaving from Education and Training and Related Matters through the Lens of the Life Course Paradigm: A Systematic Review of the Literature

Laura Guerrero-Puerta [1] and Mónica Torres Sánchez [2,*]

1    Departmento de Didactica, Organización Escolar y Didácticas Especiales, Universidad Nacional de Educación a Distancia (UNED), 18071 Madrid, Spain; laura.guerrero.puerta@edu.uned.es
2    Departamento de Teoría e Historia de la Educación y M.I.D.E., Universidad de Málaga, 29010 Málaga, Spain
*    Correspondence: motorres@uma.es

**Abstract:** Here, we present a systematic review of the literature on Early Leaving from Education and Training (ELET), which uses the life course paradigm as an explanatory model or approach. This review has returned little in the way of scientific literature, although interest in the topic has been growing in recent years, which addresses the ELET process from different points of view. First, we highlight the means that this review provides to contextualize ELET in relation to new age-specific norms, reflecting on the process that has led to it. In addition, this review suggests that it is increasingly important to change the focus of research on ELET, exploring the process within a framework of complex trajectories, including the possibility of returning once ELET has occurred.

**Keywords:** Early Leaving from Education and Training; school failure; dropout; life course research; youth; school-to-work transitions

## 1. Introduction

Over the past few decades, school enrolment rates have risen in all developed geographical regions. This trend can be attributed in part to significant changes in the conditions affecting the transition from school to work for young individuals (Parreira do Amaral et al. 2015). Additionally, a coexistence of devaluing basic education qualifications and a remarkable emphasis on credentials in society have played a role in this increase (Castro 2010; Fernández 1998).

In this context, leaving education or training prematurely portends an uncertain future for those who make such a decision, as numerous studies have established correlations between Early Leaving from Education and Training (hereafter ELET) and various aspects, including poorer health and limited civic participation within this group (Vanttaja and Järvinen 2006; Acevedo et al. 2015; Orepoulos 2007). Furthermore, ELET not only poses risks from an individual perspective but also has structural implications, leading to a higher probability of unemployment, precarious job opportunities, and reliance on social assistance for those experiencing this process (NESSE 2009). This, in turn, can hinder national growth due to increased public expenditure, reduced minimum wages, and elevated unemployment rates, among other factors (Psacharopoulos 2007; Rouse 2007).

Consequently, many countries have prioritized efforts to reduce ELET to foster economic growth and social cohesion among their citizens. Various strategies and organizations, such as the European Union and its 2010 and 2020 strategies, or the United States with its National Dropout Prevention Center and dropout prevention month, are dedicated to achieving this objective through personalized approaches to increasing graduation rates.

While the scientific literature on ELET is extensive, classical studies have often overlooked the interactive nature of the determinants (EASNIE 2016) and, at times, proposed causal relationships that cannot be solely explained by ELET. Early Leaving from Education

or Training is a complex process that may have roots even before formal schooling begins. It involves multiple interacting risks and protective factors operating at different levels that impact young individuals (Martinez et al. 2004). These factors, traditionally categorized as individual characteristics, family, school, and social conditions, are influenced by broader socio-historical and economic contexts (EASNIE 2016). To effectively inform policy decisions aimed at reducing ELET rates, the scientific literature must go beyond merely identifying factors or consequences. Instead, comprehensive practices and policies should be developed, addressing ELET holistically (EASNIE 2016).

This research aims to address this need and proposes the life course explanatory model of school leaving to enhance understanding of the socio-historical conditions shaping ELET. Through a literature review, we explore existing studies that utilize the life course paradigm as both an explanatory model and an approach to this process. Accordingly, our study seeks to answer two fundamental questions:

(a) What aspects of ELET have been examined through the life course explanatory model?
(b) What further questions does this research raise?

## 2. Theoretical Framework

Below, we present a comprehensive exploration of ELET using various theoretical models found in the literature. Subsequently, we critically assess the suitability of employing the life course approach to explore ESL. But to gain a better understanding of what ELET means and what concepts are related to it, we start this section by briefly discussing the conceptualization of the term.

### 2.1. Early Leaving from Education and Training and Associated Terms

The concept of Early Leaving from Education and Training in the literature stands out due to its ambiguous and polysemic nature (Tarabini et al. 2015; Tarabini and Rambla 2015; NESSE 2009). Depending on the context and evaluating entity, variations can be observed in terms of age, education level, and job readiness (Guerrero-Puerta 2022; Guerrero-Puerta and Guerrero 2021). The definition of ELET varies among international organizations such as Eurostat and the OECD. Eurostat has recently introduced the term, "Early Leaving from Education and Training" to enhance the definition and include alternative pathways to the concept of Early School Leaving, which previously only considered educational routes. This term refers to individuals aged 18 to 24 with qualifications below upper-secondary education level, who are not engaged in education or training. On the other hand, the OECD refers to individuals aged 15 to 24 who are not in education or training and have not completed basic secondary education (OECD/EU 2018).

Given this diversity of terms, the literature points out that, despite efforts to achieve comparability in the definition of ELET, there are numerous divergent factors in its calculation depending on the region evaluating it. This hinders the uniformity and precision of the term (Fernández-Macías et al. 2010; García Fernández 2016). The variety in numerical criteria and certifications across different European countries highlights the lack of specificity. Additionally, it is important to mention that in most cases, when referring to ELET, minors are not encompassed, creating a gap in categorizing individuals aged 16 to 18 who leave before reaching the legal age of completing studies in a significant number of European countries. Furthermore, historically, the term, ELET is linked to other terms such as absenteeism, school failure, or dropout, intensifying the ambiguity of its definition and complicating its coherent application in the educational sphere.

These terms, representing a historical evolution of the understanding of the act of leaving, substantially differ based on the nuances and connotations brought by the words used. For instance, school failure refers to not meeting educational objectives, while school dropout can involve voluntary withdrawal or "drop" without acknowledging the structural factors (Fernández-Macías et al. 2010; Miñaca and Hervás 2013). These ambiguities and differences further complicate the comprehension and approach to ELET in the educational

system (Guerrero-Puerta 2022). Therefore, in this article, while analyzing the literature, the decision has been made to respect the terminology used by the authors.

### 2.2. Early Leaving from Education and Training and Its Portrayal in the Literature

The literature has identified numerous factors that partially explain the departure from education or formation at an early age, reaching a consensus about how this is a multi-causal and gradual process that cannot be attributed solely to isolated factors. As a result, researchers have developed explanatory models that attempt to elucidate the complexity of this process by examining the interplay of factors associated with the process. For this section, we draw upon the studies conducted by Rumberger and Lim (2008), Benítez-Zabala (2016), and Guerrero-Puerta (2022), which have revealed a total of eight models. Each of these models will be explored in the following section.

(A)   Tinto's explanatory model of dropout

Tinto's model (Tinto 1987, 1994) emphasizes the role of the institutional environment in influencing students' integration and, consequently, their decision to drop out (Benítez-Zabala 2016). Initially, personal attributes, such as family background, skills, knowledge, and previous school experiences, predispose the dropout process. Once students begin schooling, other factors come into play, including social integration and the perceived value of education, as well as academic integration within the educational environment. Both social and academic dimensions are influenced by the formal structures of the educational institution, and a relatively high level of integration in both dimensions is necessary for students to continue their educational journey. External factors may also mediate this decision (Rumberger and Rotermund 2012). This model introduced a significant contribution to the literature, shifting away from an individualized approach to dropout by acknowledging the influence of the school institution's structure on the process. Additionally, Tinto's model suggested that transferring students to schools with better integration conditions could prevent dropout, recognizing the importance of external factors in the dropout process.

(B)   Finn's explanatory model of dropout

Finn (1989) proposes two complementary models to explain students' failure to complete secondary education, focusing on behavioral, academic, and psychological aspects (Rumberger and Rotermund 2012). The "self-esteem-frustration model" centers on disruptive behavior and negative attitudes towards the school environment, such as absenteeism. The "identification-participation model" highlights school participation and a sense of identification with education. While Finn's work is considered one of the first to delve into the process of school disaffection or disengagement, the more recent research (Finn and Zimmer 2012) has shifted focus from the Self-Esteem–Frustration model to the Identification–Participation model.

(C)   Explanatory model of dropout according to Wehlage

Wehlage and his colleagues' model (Wehlage et al. 1989) emerged from an empirical study and is strongly influenced by the Finn and Tinto models (Rumberger and Rotermund 2012; Mastrorilli 2016). Notably, this model introduces the concept of educational engagement as a crucial element in the psychological and cognitive aspects of the dropout process. Educational engagement refers to the characteristics that facilitate student participation in learning. Wehlage emphasizes the significance of a positive bond between educational staff and students, fostering mutual support and commitment. The model distributes responsibility for school success or failure between the school and the student (Mastrorilli 2016). According to Wehlage et al. (1989), impediments to this process include unmotivating schoolwork, abstract and non-participatory learning processes, and inadequately addressed knowledge acquisition.

(D)   Explanatory models of student dropout surrounding deviance

Rumberger and Lim (2008) notes that a significant proportion of models identified in his systematic review focus on attitudes and behaviors exhibited by students in and

out of school, including juvenile delinquency, drug and alcohol abuse, and parenting during adolescence. These "deviance" models explore behaviors that deviate from normative expectations, linking them to disengagement from school and subsequent dropout. Studies indicate a direct relationship between juvenile-justice-related court proceedings, arrests, periods of incarceration, and dropout (Ward et al. 2015; Aizer and Doyle 2015; Rud et al. 2018).

(E)   Rumberger's explanatory model of dropout

Although initially considered a conceptual framework, Rumberger's model (Rumberger and Larson 1998) has been regarded as an explanatory model of dropout in the literature. It rejects rigid causal relationships and emphasizes a complex web of circumstances shaping the decision to drop out. Rumberger identifies individual and institutional factors, which are interrelated bidirectionally, but the relationship between institutional and individual factors is only considered from the first group to the second. The model highlights that early dropout cannot be understood through isolated factors and underscores the importance of educational stability for a good level of educational attachment, considering mobility a risk in the educational process.

(F)   Tedesco's explanatory model of school failure and early school leaving

Tedesco (1983) also views school failure and early school leaving as multidimensional processes, avoiding the establishment of rigid causal relationships. Tedesco's model emphasizes the need to move away from biological and sociological determinisms, which limit intervention possibilities. Instead, he considers multiple factors that reinforce or mitigate each other's effects. He highlights the importance of considering both educational system variables and cultural differences between students. Tedesco encourages analyses that transcend structures and dynamics at different levels and identifies interactive relationships leading to exclusion in certain economic, social, and cultural contexts. Educational inclusion/exclusion requires a focus on both schools and the broader contexts impacting students' trajectories.

(G)   Escudero's explanatory model of school failure and desertion

Escudero's model, inspired by Tedesco (1983) and Rumberger and Larson (1998), views school failure and desertion within the logic of educational exclusion, providing a more holistic understanding of these process. Escudero calls for interventions addressing both the school context and the broader contextual factors that influence early school leaving and educational failure. The model explores how educational failure is a product of multiple factors, including the dynamics of economic transformation influenced by neoliberal capitalism, and how interventions must be tailored accordingly to mitigate exclusion and vulnerability. The model also questions traditional conceptualizations of failure and encourages a rethinking of basic education to ensure universal rights are upheld.

(H)   Pathways and transitions after leaving school model by GRET (Education and Work Research Group)

The model proposed by the Education and Work Research Group (GRET) examines ESL pathways and transitions after Compulsory Secondary Education, considering a broader understanding of youth transitions. GRET conceptualizes early school leaving as a process resulting from three dimensions: socio-historical, biographical-subjective, and political-institutional. The model identifies possible educational and employment pathways following ELET, ranging from returning to education to various trajectories in the labor market, each with its unique characteristics and probabilities of success. GRET emphasizes the need for qualitative research to gain a deeper understanding of individual trajectories and how local-level contexts shape these pathways.

*2.3. Theoretical and Methodological Life Course Paradigm*

In the preceding section, we explored various models developed to understand ELET and the diverse explanations associated with it. This investigation revealed an increasing

demand to transcend rigid explanatory models and explore multidimensional perspectives. Therefore, in this section, we will delve into the life course paradigm and its potential for analyzing ELET.

2.3.1. The Concept of the Life Course Paradigm

The term, "life course", has multiple meanings, encompassing both a specific concept and an entire theoretical–methodological paradigm. As a concept, life course refers to life patterns, categorized by age, that exist in societal structures and are influenced by historical transformations. (O'Rand 1998). On the other hand, as a paradigm, life course represents a comprehensive research field, relatively recent in its creation, for interpreting life trajectories (Elder et al. 2003).

For the purposes of this article, we will focus on the characteristics of the life course as a paradigm and the insights it offers for ELET research. The significance of the life course paradigm lies not only in its theoretical potency to comprehend the social structure of life courses but also in its potential as an analytical tool to gain a deeper understanding of individual trajectories over time.

2.3.2. Origins and Contributions of the Life Course Paradigm

The life course paradigm emerged in the United States during the 1970s, introduced by sociologist Glen Elder. This approach sought to distance itself from the concept of a fixed life span, which presented life as a linear and normative progression tied to age and developmental stages (Kovacheva et al. 2015). It also sought to depart from perspectives that primarily focused on biological development in the life cycle (Elder et al. 2003).

In contrast, the life course paradigm proposed that life consists of graded transitions through social positions and structures embedded in relationships that influence behavior. It acknowledged the interconnectivity between individual developmental trajectories and the lives of others (Elder 1998). This innovative analytical approach provided a more nuanced and holistic perspective compared to its predecessors. By emphasizing the significance of individual agency in shaping life courses, it also recognized the importance of understanding the relationship between individual lives or agency and social change, with a focus on studying the social, economic, and historical forces that influence them (Blanco and Pacheco 2003).

As mentioned earlier, the life course paradigm possesses substantial theoretical potential for comprehending the social structure within life courses and serves as an analytical tool to study individual trajectories throughout life. Its ambitious intention lies in providing an analytical framework to explore the interactions and intersections between the micro-level of individuals and the macro-levels of culture, economy, and social policy. To grasp how life course analysis operates, it is crucial to consider the five fundamental life course principles (Elder et al. 2003).

2.3.3. Fundamental Life Course Principles

1.  The Principle of Lifespan Development: This principle emphasizes the need for a long-term perspective in investigating or analyzing life courses as a cumulative process of life experiences (Elder et al. 2003).
2.  Agency: Life course theory acknowledges that individuals determine their life course within certain constraints and opportunities. Agency is inevitably linked to historical and social forces, allowing people to shape their lives within specific conditions (Elder et al. 2003).
3.  Time and Place: This principle posits that individuals belonging to specific birth cohorts are heavily influenced by historical context and location. Each birth cohort faces unique constraints and opportunities that shape their life trajectories. To fully understand individual behavior and decisions, we must consider the effectiveness and direction of past or future individual life plans (Hareven 1994), as agency is

influenced not only by present situations but also by the "shadows of the future" and the "shadows of the past" (Bernardi et al. 2019).

4.  Timing: The impact of individual experiences and historical events on subsequent life courses depends significantly on the lifetime in which they occur. This principle considers the intersection between individual time (age), family time (stage of the family cycle), and historical time (economic cycles, social changes).
5.  Linked Lives: This principle underscores the interdependence of life courses with each other.

It is essential to recognize that these principles operate bidirectionally and interdependently, creating intricate relationships. A thorough examination of these principles, along with their complex interdependencies, is crucial to understanding contemporary life courses (Bernardi et al. 2019). Consequently, this understanding is also key to advancing our comprehension of the ELET process.

The life course paradigm lacks a single, all-encompassing theory, leading researchers in this field to refer to it as a paradigm or approach. The ecumenical nature of these principles, not excluding stricter theoretical approaches, makes it identifiable as a paradigm rather than a theory. This flexibility limits the analytical scope of the life course since it does not offer an explicit analytical framework. We position the life course paradigm as a heuristic device for examining the interactions between individual lives and social change, as well as for studying the interrelationship of social trajectories and trajectories of societal development and change (Tikkanen et al. 2021).

Life course studies often concentrate on individual life courses and how they are affected by macro-level social changes. Researchers explore how different institutions play a role in shaping individual opportunities and decision-making processes. Based on Marshall and Mueller's (2003) synthesis of Walther Heinz's efforts in delimiting life course research, studies within this paradigm can be categorized into three approaches: cohort approaches focusing on social change and transitions across generations and cohorts, constructionist approaches highlighting agency, biography, and personal narratives, and institutional approaches examining the interaction of individuals and policies concerning transitions.

Regarding methodology, there is a clear gap in life course research. Blanco (2011) highlights two main trends: studies with a mixed-methodology emphasis that combine qualitative investigation of individual lives with statistical data and purely qualitative research that explores life stories through semi-structured interviews. Notably, this paradigm's initial design relied on quantitative data for longitudinal studies. However, how can we connect this to the study of ELET?

*2.4. Studying Early Leaving from Education and Training from a Life Course Paradigm Approach*

Life course studies have traditionally centered on understanding how individual life courses are affected by macro-level societal changes and the influence of different institutions on individual opportunities and decision-making processes (Mills 2007). Consequently, significant research has focused on studying the impact of these social changes on the trajectories and transitions of young people. To explore how applying the life course paradigm can advance ELET research, it is essential to investigate progress in using this paradigm in youth studies.

Studies in education and youth using the life course perspective have emphasized the de-standardization of life courses due to rapid social developments triggered by factors such as changes in welfare state provision systems, increasing credentialism, changes in family structures, and fluctuations in the labor market (Brückner and Mayer 2005). This de-standardization has resulted in youth experiencing uncertainty, vulnerability, and reversibility, departing from the previously linear and gender-differentiated path to adulthood (Bynner 2016; Walther 2015).

While the nature of the life course has evolved, educational and youth policies persist in being designed around a linear and predictable process associated with certain "rites of passage" to adulthood. As a result, youth policies often rely on age as a primary criterion,

leading to tensions as young people navigate life courses typical of past linear generations (Benasso 2015). This position is often interconnected with lifelong learning and/or employability discourses, from which an assumed equality of opportunities is provided by education/training processes, based on new models that emphasize "economized" discourses surrounding human capital (Tarabini and Jacovkis 2021). Ross and Leathwood (2013) argue that this is due to a simplistic understanding of educational pathways, as well as the school-to-work transition processes that have evolved in contemporary societies as part of a limited comprehension of the youth.

Acknowledging these changes in the youth segment, a broader approach to ELET is needed. Such an approach should integrate socio-historical factors, explore new patterns of youth transition, and investigate the factors contributing to educational/training trajectory exclusion. In light of this, the proposal made by Abiétar and Torres (2018) becomes relevant as it advocates addressing ELET from the life course paradigm. This approach integrates the process within a historical context, strongly linked to globalization and the development of neoliberal policies that have influenced labor market changes and subsequently the life course of young people. It incorporates a biographical perspective to explore the dialectical relationship between life courses and the biographies of young individuals.

Thus, the life course paradigm can tackle four significant challenges that are currently underexplored in ELET related literature: (1) questioning the existence of "normal and standardised" life trajectories contributing to the development of normative expectations, leading to structural inequality issues that affect vulnerable groups, including early school leavers; (2) reevaluating specific programs and policies aimed at reducing ELET and failure, examining whether they recognize the heterogeneous life projects of young people or adhere to linear and standardized projections; (3) promoting training and educational experiences that offer stability for young individuals in an era characterized by precariousness, and (4) exploring models that recognize both formal and informal learning competencies to facilitate flexible transitions between education, training, and employment.

The life course paradigm presents significant potential for understanding ELET, although it remains open to multiple interpretations through diverse methodological and theoretical approaches. To gauge the extent of its application, it is essential to conduct a systematic review of the literature on ELET to assess the current state of the field. This research question calls for further exploration in the subsequent sections.

## 3. Methodology

Due to the extensive scientific production on the topic of ELET, it is crucial to conduct literature reviews that systematically delimit different themes within this line of research and its achievements. Therefore, we have adopted a systematic literature review approach for this article, aiming to distance ourselves from less exhaustive literature reviews that may present researcher-induced biases, such as narrative literature reviews.

Conducting a systematic literature review involves employing strategies to minimize bias by integrating, critically analyzing, and synthesizing the entire body of literature available on a given topic. This approach focuses on specific research questions, employs a clearly defined search strategy, sets specific selection criteria, and entails rigorous and critical analysis of the information (Sánchez Meca 2010). The systematic review strives for a more comprehensive analysis of the literature compared to narrative reviews, seeking to encompass the totality of research published on a specific topic (Fernández-Ríos and Buela-Casal 2009). In this context, our systematic review takes a concrete approach, examining the literature on ELET with a focus on the life course paradigm. The process followed to carry out this systematic review is outlined below.

*The Systematic Review Process*

The literature review was conducted, utilizing Web of Science (WOS) and Scopus as the primary search engines. To ensure the systematic nature of the process, the term, "Abandono escolar" (early school leaving) was introduced in the UNESCO Thesaurus,

which returned related terms, including "Deserción escolar" (school dropout), "Fracaso escolar" (academic failure), "drop out," "School leaving,", and "Academic Failure."

Inclusion Criteria:

1. Theoretical and empirical scientific articles published in journals indexed in the search engines described.
2. Articles that analyze ELET, considering a theoretical and/or methodological approach of the life course paradigm or utilize its theoretical tools.
3. Articles written in English or Spanish.
4. Articles published between 1990 and 2018.

Sample:

After defining the inclusion criteria and research objectives, an initial literature search yielded a population of 157 articles. These articles were then analyzed based on the proposed inclusion criteria. After removing duplicate articles and those not aligned with the research scope, a final sample of 17 articles was obtained. The workflow is depicted in Figure 1 below:

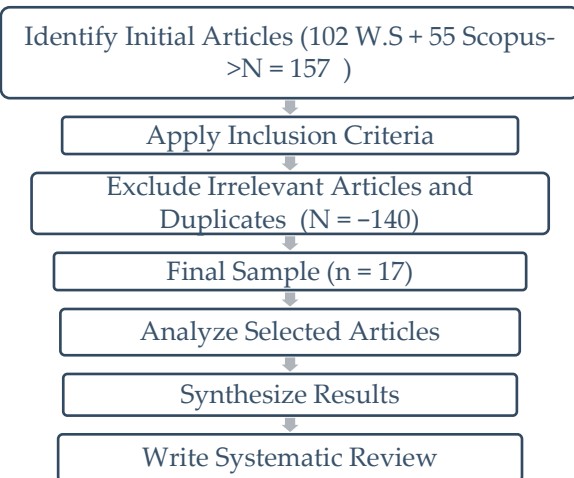

**Figure 1.** PRISMA Diagram. Source: Own Elaboration.

## 4. Results: Elet from a Life Course Perspective

The present section presents the findings of the systematic review, offering significant insights into the research landscape concerning ELET from a life course perspective. The section starts with a bibliometric analysis, followed by a summary of the content included in the selected articles. Lastly, the section delves into the emerging lines of research pertaining to ELET within the life course paradigm.

### 4.1. Bibliometric Analysis

After conducting the necessary screening and systematization, a total of 17 articles were identified that adopt the life course perspective to explore the process of ELET. Table 1 in this section presents the essential information of these papers, providing bibliometric insights into the systematic review focused on ELET within the framework of the life course theoretical–methodological paradigm.

Additionally, this section will include an in-depth analysis of the gathered information (see Table 1). Through this comprehensive analysis, our aim is to shed light on the characteristic features and prevailing trends observed in the selected papers, thereby offering valuable insights into the research landscape surrounding this critical topic. By scrutinizing aspects such as the distribution of paper types, specific fields of study, temporal patterns, and representation across languages, we aspire to develop a deeper understanding of the academic interest, thematic focus, and global reach of research endeavors related to ELET

and its interconnectedness with the life course paradigm. This examination of the literature will facilitate a more informed and nuanced discussion and interpretation of the findings in the subsequent sections.

**Table 1.** Papers Included in the systematic review.

| Authors | Year | Title | Type | Field |
|---|---|---|---|---|
| Alexander, Entwisle and Kabbani | 2001 | The Dropout Process in Life Course Perspective: Early Risk Factors at Home and School | Empiric | Economic |
| Chen and Kaplan | 2003 | School Failure in Early Adolescence and Status Attainment in Middle Adulthood: a Longitudinal Study | Empiric | Sociology |
| Entwisle, Alexander, and Olson | 2004 | Temporary as Compared to Permanent High School Dropout | Empiric | Sociology |
| Entwisle, Alexander, and Olson | 2005 | Urban Weenagers. Work and Dropout | Empiric | Sociology |
| Pallas | 2006 | A Subjective Approach to Schooling and the Transition to Adulthood | Empiric | Sociology |
| Bersani and Chapple | 2007 | School Failure as an Adolescent Turning Point | Empiric | Sociology |
| Janosz, Archambault, Morizot, and Pagani | 2008 | School Engagement Trajectories and Their Differential Predictive Relations to Dropout. | Empiric | Psychology |
| Ruseell Rumberger and Susan Rotermund | 2012 | The Relationship between Engagement and High School Dropout | Revision | Education |
| Schmid and Stalder | 2012 | Dropping out from Apprenticeship Training as an Opportunity for Change | Empiric | Sociology |
| Minguez | 2013 | The Early School Leaving in Europe: Approaching Explanatory Factors | Empiric | Education |
| Pharris-Ciurej, Hirschman and Willhoft | 2012 | The Ninth Grade Shock and the High School Dropout Crisis | Empiric | Sociology |
| Ríos | 2014 | La Cohorte Pisa 20016-2011 en Uruguay | Empiric | Sociology |
| Gibbs and Heaton | 2014 | Drop out from Primary to Secondary School in México: a Life Course Perspective | Empiric | Psychology |
| Dupéré, Leventhal, Dion, Crosnoe, Archambault and Janosz | 2015 | Stressors and Turning Points in High School and Dropout: a Stress Process, Life Course Framework | Revision | Education |
| Campbell | 2015 | High School Dropouts after They Exit School: Challenges and Directions for Sociological Research | Empiric | Sociology |
| Vogt | 2017 | Age Norms and Early School Leaving | Revision | Sociology |
| Holtmann, Menze, and Solga | 2017 | Persistent Disadvantages or New Opportunities? The Role of Agency and Structural Constraints for Low-Achieving Adolescents' School-to-Work Transitions | Empiric | Sociology |

Source. Own Elaboration.

The distribution of paper types reveals that empirical studies predominate, accounting for approximately 83% of the selected papers, while revision papers constitute the remaining 17%. This indicates a tendency on conducting empirical research to understand ELET and related matters within the context of the life course paradigm.

In terms of the specific fields, sociology emerges as the most prominent with eleven papers 64.7%), followed by education with three papers (17.6%), psychology with two papers (11.7%) and economics with one (5.9%). This distribution suggests that the study of this subject from a life course perspective is more common within the sociological domain.

Analysing the temporal distribution, it is evident that the majority of the papers were published after 2010, with 10 papers falling into this period (58.8%). This may indicate a growing interest in the topic in recent years and suggests an increasing focus on exploring ELET through the lens of the life course paradigm. One noteworthy observation from the bibliometric analysis is the very limited production of research in Spanish. Among the seventeen identified articles, only one was published in Spanish (5.9%), while the remaining sixteen were written in English and published in Anglo-Saxon journals. This disparity highlights the prevalence of English as the dominant language for scholarly communication in this field, potentially limiting the accessibility and dissemination of research findings to Spanish-speaking communities.

In summary, the analysis of bibliometric information reveals a significant academic interest in investigating ELET from a life course theoretical-methodological paradigm. The predominance of empirical studies and the growing number of publications in recent years indicate a strong commitment to comprehensively understanding the factors influencing ELET and its long-term implications on individuals' life trajectories. The interdisciplinary nature of the research, involving fields like sociology, psychology, and education, demonstrates the complexity and multi-dimensional approach adopted to address this crucial issue in educational research. However, the limited representation of Spanish-language publications underscores the need for greater diversity and inclusivity in academic publishing to ensure broader dissemination of research outcomes and perspectives.

*4.2. Sumary of Content in Articles Included in the Review*

Following the bibliometric analysis, an essential component of this review is the overview of content, provided in Table A1 (In Appendix A). This summary presents valuable insights into the main themes, findings, and implications derived from each study, contributing significantly to a deeper understanding of the current state of research on ELET within the life course paradigm.

Summarizing the findings presented in this systematic review, the content analysis of the included articles provides a comprehensive compilation of studies that delve into the intricate process of leaving school. These studies employ a wide array of methodologies, ranging from individual-level analyses to comparative approaches across diverse countries and regions. The review highlights the significant contributions of various authors in shedding light on the multifaceted nature of ELET and its underlying determinants.

Ríos (2014) and Gibbs and Heaton (2014) contribute to the understanding of ELET rates and causal factors by adopting a comparative approach across European countries. These studies emphasize the pivotal role of social spending on education and education policies, underscoring their influence on ELET outcomes. In contrast, Dupéré et al. (2015), Janosz et al. (2008), and Campbell (2015) take an individual-level approach, meticulously examining educational trajectories, school engagement, and early school failure within specific student cohorts. Their work highlights the crucial significance of individual experiences and events in shaping the decision to leave education, including educational risk events and trajectories of school engagement.

Across the studies, several factors influencing ELET emerge, encompassing elements such as family background, school engagement, socioeconomic conditions, and personal resources. Ríos (2014) points out the role of inequalities related to social risks on academic events, highlighting how normative age milestones impact educational trajectories. Dupéré

et al. (2015) propose a stress process model that integrates long-term vulnerabilities and immediate events, challenging traditional dropout perspectives. Janosz et al. (2008) identify distinct trajectories of school engagement and their predictive associations with dropout, advocating for targeted interventions. Entwisle et al. (2005) explore the interplay between employment types and dropout risk, revealing nuanced patterns based on age and job type. Schmid and Stalder (2012) delve into dropout from vocational education and training programs, indicating that leaving can lead to positive realignments for specific individuals.

Collectively, the studies underscore the intricate nature of the dropout process, shaped by a multitude of individual, contextual, and structural factors. Rumberger and Rotermund (2012) provide an overview of prominent dropout models, emphasizing school-engagement-related factors. Mínguez (2013) adopts a comparative approach to analyze early school leaving rates across European countries, showcasing the impact of social spending on education and policies. Campbell (2015) not only discusses reasons for dropout but also emphasizes resulting consequences, advocating for comprehensive research to unravel the mechanisms linking dropout to unfavorable outcomes.

Furthermore, several studies delve into the longitudinal impact of dropout on an individual's trajectory. Chen and Kaplan (2003) examine how early school failure influences status attainment in midlife, considering mediating factors such as mental health and deviant behaviors. Pallas (2006) challenges traditional perspectives on adulthood transitions. Alexander et al. (2001) analyze risk factors and resources at different schooling milestones, demonstrating their cumulative and interactive effects on dropout risk. Holtmann et al. (2017) highlight challenges faced by low-achieving individuals transitioning to the workforce, emphasizing personal agency and structural constraints.

Approaching ELET, Vogt (2017) draws attention to the influence of age norms on school-to-work transitions and perceptions of ELET as a form of deviance. Chen and Kaplan (2003) also adopt a life course perspective, examining the long-term implications of early school failure, identifying mediating pathways beyond years of education, such as mental health and deviant behaviors.

Moreover, the systematic review addresses policy implications derived from multiple studies. Ríos (2014) advocates for policies integrating labor and educational trajectories to effectively address educational risks stemming from social inequalities. Mínguez (2013) underscores the urgent need for European welfare states to fortify education policies and enact comprehensive reforms to reduce ELET effectively. Campbell (2015) emphasizes the importance of future research in comprehending the multifaceted consequences of ELET, encouraging consideration of factors extending beyond labor market penalties.

Overall, the contrasting perspectives presented by the authors collectively contribute to a holistic understanding of ELET. The studies highlight the importance of early resources, critical events, socioeconomic factors, age norms, and vocational contexts in shaping ELET outcomes. By considering these diverse viewpoints, policymakers and educators can develop more targeted interventions to prevent ELET and promote successful educational trajectories for all students.

*4.3. ELET and Related Matters through Life Course Lenses—Different Approaches in Literature*

Given the specific focus of this article on the life course paradigm, it becomes imperative to delve into the various approaches that this paradigm has garnered within the selected literature. Table A2 (In Appendix B) provides a succinct overview of the varied approaches and findings within each selected study, showcasing how the life course paradigm is applied to understand the process of Early Leaving from Education and Training. The table encapsulates the range of perspectives and insights garnered from these studies, contributing to a comprehensive understanding of ELET within the context of individuals' life course trajectories.

When comparing and contrasting the studies summarized in the literature review and their utilization of the life course perspective, several consistent approaches emerge. Across these studies, there is a common thread that underscores the interplay between

individual agency and structural constraints in shaping life trajectories. This dynamic interaction is observed in works like Ríos (2014), Gibbs and Heaton (2014), Dupéré et al. (2015), Janosz et al. (2008), and Entwisle et al. (2004). These researchers acknowledge that personal decisions and actions are intricately intertwined with broader societal, economic, and educational contexts. This recognition resonates with the core tenets of the life course paradigm, which highlights the importance of comprehending both individual agency and external influences when deciphering life paths.

Several studies, such as Gibbs and Heaton (2014), Chen and Kaplan (2003), Vogt (2017), and Pallas (2006), delve into the realm of longitudinal analysis and trajectories. These inquiries involve tracing the impact of experiences and events across various life stages on educational outcomes and transitions. This exploration aligns closely with the life course paradigm's central focus on understanding the shifts and milestones that occur over time within the context of historical, social, and individual circumstances.

Transitions and turning points are core concepts in studies like Bersani and Chappie (2007) and Entwisle et al. (2004). These works emphasize how specific life events or pivotal changes can drastically alter an individual's trajectory. This perspective harmonizes with the life course paradigm's emphasis on comprehending life transitions as decisive moments capable of catalyzing significant shifts in behavior and consequences.

In line with the holistic approach of the life course paradigm, numerous studies recognize the intricate interplay of diverse factors influencing educational outcomes or transitions. Alexander et al. (2001), Pallas (2006), and Pharris-Ciurej et al. (2012) exemplify this approach. These investigations emphasize the necessity of accounting for a wide array of factors—ranging from individual traits to familial dynamics and broader societal conditions—to obtain a comprehensive understanding of educational pathways.

Moreover, certain studies, such as Rumberger and Rotermund (2012) and Schmid and Stalder (2012), emphasize the importance of contextualizing educational experiences within the broader scope of an individual's life. They illuminate how leaving education can lead to various trajectories, including reentry or shifts in career directions. This emphasis aligns with the life course paradigm's emphasis on situating individual development within historical and societal contexts.

Finally, the influence of labeling and social norms on educational outcomes and life trajectories is explored in studies like Vogt (2017) and Pallas (2006). These works illustrate how societal expectations and norms can significantly shape individuals' decisions and behaviors, aligning with the life course paradigm's recognition of the intricate interplay between personal lives and broader societal changes.

Collectively, the studies underscore how the life course perspective is effectively employed to decode educational outcomes, transitions, and trajectories. They underscore the dynamic nature of individual development, the intricate interplay of myriad factors, and the necessity of considering historical, social, and individual contexts to truly fathom educational pathways.

### 4.4. Early Leaving from Education and Training from a Life Course Perspective—Emerging Lines of Reseach

Despite offering a comprehensive summary of the content covered in the selected articles, this systematic review uncovers several compelling topics that demand further investigation and exploration. The main advancements presented in this research allow us to gain a better understanding of the current state of the art in ELET research from this paradigm and identify the key research areas that have emerged as a result. By delving deeper into these topics, researchers can advance our knowledge and contribute to a more comprehensive understanding of Early Leaving from Education and Training, thereby informing targeted interventions and shaping future directions for ELET research.

A.    Early Leaving from Education and Training as an age-specific norm

The development of studies focusing on the life course paradigm has provided valuable insights into youth transitions and their de-standardization process, challenging the

notion of "normal and standardized" life trajectories and its implications on ELET and school failure (Abiétar and Torres 2018). One prominent line of research in the literature on ELET gravitates towards conceptualizing it as an "age-specific norm".

Scholars such as Vogt (2017), Janosz et al. (2008), and Ríos (2014) argue that the prevailing concept of ESL is closely tied to age-specific criteria, which engender expectations about educational attainment at different life stages. Consequently, those who fail to meet these age-specific norms, by not completing secondary or tertiary education within the prescribed period, find themselves outside the boundaries of normativity (Vogt 2017).

According to Vogt (2017), the emergence of the institutionalized indicator of ELET has further solidified age-specific norms in education. This shift has been motivated by various states' focus on producing students with basic education to cope with modern social requirements, where higher education levels are increasingly demanded, and the adaptability to rapid societal changes is essential (Janosz et al. 2008). Vogt (2017) highlights the case of the European Union, which, in response to the crisis, has emphasized ELET rates as a constraint and promoted education and training as a universal solution to boost economic growth.

Consequently, exploring the impact of internalizing failure due to non-attainment of secondary education credentials and the challenges faced in a labour market that increasingly demands such qualifications (Janosz et al. 2008) becomes crucial. Nevertheless, Vogt (2017) notes that the study of ELET, particularly questioning age-specific norms, remains relatively unexplored in the literature.

B.     ELET and the need to explore it as part of broad trajectories and complex transitions

Many studies adopt the concept of trajectory to examine ELET, emphasizing that it is part of a broader path that can manifest early in a student's schooling and extend beyond the educational sphere (Janosz et al. 2008; Ríos 2014; Gibbs and Heaton 2014; Rumberger and Rotermund 2012; Hirschfield 2009; Chen and Kaplan 2003). However, most continue to view ELET as an endpoint in the educational/training realm without adequately exploring the potential for re-entry at other stages of life.

Dupéré et al. (2015) challenge this static view of early school leaving as a complete rupture with education, arguing for a more dynamic perspective. Entwisle et al. (2005) similarly criticize the conventional perception of the school-to-work transition as absolute, calling for greater attention to the complexities of transitions mentioned in the theoretical framework.

Vogt (2017) provides an intriguing example of how this conception permeates the calculation models used to assess ELET rates, making it challenging to differentiate between true school leavers and individuals temporarily pausing their education. This raise concerns that inadvertently downplaying alternative paths after leaving might limit attention to creating supportive routes for re-entry. The issue prompts questions about post-ELET experiences and strategies to facilitate the return of those who have left the education system.

## 5. Discussion

In the examination of various models, the challenge of Early Leaving from Education and Training emerges as a concern that questions the reliability of educational institutions. These institutions must strike a balance between accommodating a diverse student population while ensuring equitable access to basic education for all (Tedesco 1983; Escudero Muñoz 2005). To understand the origins and persistence of this process, engagement with theoretical frameworks becomes imperative. However, the adoption of these frameworks must proceed cautiously, avoiding inflexible constructs, as underscored by Benítez-Zabala (2016), who points out gaps in the literature regarding the process that leads to leave school early.

Recent socioeconomic changes have significantly altered the youth experience, leading to non-standardized trajectories involving transitions across various life domains. Consequently, the educational and training realm demands a broader understanding of the

intricate interplay between work and school transitions, necessitating a reevaluation of the concept of ELET within this evolving context (Abiétar and Torres 2018). In this pursuit, the proposal to embrace a life course perspective emerges—not as a rigid and static model, but as a versatile framework encompassing the socio-historical shifts that influence the life stories of young individuals (Blanco 2011; Bernardi et al. 2019). This approach inherently encourages a multifaceted exploration of social pathways, developmental routes, and transformative societal dynamics.

Within this intellectual landscape, a systematic review of ELET and related matters literature from a life course perspective illuminates the current state of research and identifies areas for advancement. The body of scholarly output shows growth and offers diverse viewpoints on the process of leaving education or formation early, occasionally presenting challenges when categorizing the discourse. The review identifies two crucial lines of research. Firstly, contextualizing ELET within new age-specific norms provides insights into the impact of prevailing discourses on those who leave (Vogt 2017). Secondly, emphasizing ELET within a framework of complex trajectories, including the possibility of re-entry, necessitates qualitative and quantitative studies for deeper insights. These future research directions enhance our comprehension of ELET from a life course perspective, contributing to well-informed interventions and policies.

Moreover, the theoretical framework also indicates that concerning the act of abandoning education or training, researchers have devised various explanatory models to shed light on the complex interplay of factors contributing to this multi-causal and gradual process. While each model offers valuable insights, adopting a life course perspective presents distinct advantages over these individual models. A comparison of the life course perspective with these models highlights the comprehensive and nuanced understanding that the life course perspective can provide.

Tinto's model, which accentuates the role of the institutional environment, underscores the significance of social and academic integration. Although this model acknowledges external factors, it predominantly centers on the internal dynamics and interactions within the educational institution, possibly overlooking the broader societal influences contributing to ELET (Benítez-Zabala 2016). In contrast, the life course perspective's emphasis on time and place encourages researchers to delve into historical contexts and broader societal changes that might affect educational choices, offering a more holistic understanding (Hareven 1994).

Finn's model alternates between two approaches, focusing on behavioral and psychological aspects of disengagement. While it acknowledges the complexity of students' experiences, it remains somewhat dichotomous in its approach, potentially oversimplifying the intricate interplay of factors leading to ELET (Rumberger and Rotermund 2012). Conversely, the life course paradigm's Agency principle recognizes the multifaceted nature of agency, considering the impact of historical and social forces on individual decisions, offering a more contextualized exploration (Elder et al. 2003).

Wehlage's model introduces the concept of educational engagement, which is crucial for the dropout process. However, it largely focuses on the dynamics within the educational environment and the relationships between students and staff (Mastrorilli 2016). In contrast, the life course perspective's Linked Lives principle delves into the interconnectedness of individuals' trajectories, prompting an exploration of the influence of broader social networks and communities on educational decisions, expanding the scope of analysis (Elder et al. 2003).

Deviance models link dropout to behaviors such as delinquency, drug abuse, and parenting during adolescence. While these models offer insight into behavioral aspects, they may not fully account for structural inequalities contributing to these behaviors and subsequent ELET (Rumberger and Lim 2008). The life course paradigm, with its emphasis on the interconnectedness of individual trajectories and social change, provides a more holistic understanding of how these behaviors are shaped by broader life contexts and historical factors (Elder et al. 2003).

Rumberger's model acknowledges the complexity of dropout by considering both individual and institutional factors. However, it primarily examines the relationship between these factors unidirectionally, limiting the understanding of the bidirectional influence (Rumberger and Larson 1998). The life course perspective's holistic approach, considering the bidirectional interdependencies of individual trajectories and social structures, provides a more dynamic understanding of the dropout decision (Elder et al. 2003).

Tedesco's model emphasizes multiple interacting factors contributing to school failure. While it underscores the need to transcend deterministic approaches, it might not fully account for how broader societal shifts influence these interactions (Tedesco 1983). In contrast, the life course paradigm's emphasis on Time and Place encourages the exploration of historical and contextual influences, enhancing the understanding of how these factors interact over time (Hareven 1994).

Escudero's model views school failure within the context of educational exclusion and societal dynamics. While this approach provides a comprehensive perspective, it may not fully delve into the individual agency within this complex interplay (Escudero Muñoz et al. 2009). The life course paradigm's Agency principle, combined with its consideration of broader societal changes, offers a more nuanced understanding of how individuals navigate within systems of exclusion and vulnerability (Elder et al. 2003).

The Pathways and Transitions model by GRET expands the understanding of youth transitions after ELET, considering various pathways. However, it primarily focuses on quantitative understanding and local contexts, potentially missing the broader historical influences shaping these trajectories (García Gracia et al. 2013). The life course perspective, with its emphasis on multiple principles including Time and Place, Agency, and Linked Lives, integrates both qualitative and quantitative analysis to provide a more comprehensive understanding of how historical, individual, and societal factors converge (Elder et al. 2003).

Having explored this information, and after conducting the systematic review, can be observed that in contrast to these models, the life course perspective amplifies their strengths while addressing their limitations. By accommodating individual life trajectories, historical context, agency, and social interdependencies, the life course perspective facilitates a more nuanced and comprehensive comprehension of Early Leaving from Education and Training, thereby enabling the formulation of effective policy interventions and strategies to ameliorate educational disparities. This perspective is well-suited for unraveling the complex web of factors contributing to ELET and aligns with the aim of promoting social justice and equitable educational opportunities for all individuals.

To further emphasize this affirmation, it is necessary to reinforce the notion of how the integration of the life course perspective could enhance the understanding of ELET, an area that remains scarce in the literature. The life course paradigm presents an intriguing framework for delving into the complexities of ELET offering a multifaceted lens through which to comprehend the underlying dynamics of this process. Rooted in the interplay between historical context, individual agency, and social structures, this paradigm lends itself to a nuanced analysis of the factors contributing to ELET, along with potential strategies for addressing the issue.

The advancement in the research on ELET could potentially intersects with several key principles of the life course paradigm. The principle of Lifespan Development emphasizes the importance of considering a cumulative range of life experiences over time (Elder et al. 2003). Applying this principle to ELET research involves examining how early life circumstances shape subsequent educational trajectories. By exploring the role of socioeconomic backgrounds and early educational experiences, researchers gain insights into the roots of educational disparities and their potential influence on the decision to leave education early.

Central to the paradigm, the principle of Agency recognizes that individuals make choices within specific constraints and opportunities (Elder et al. 2003). This notion invites scrutiny of the systemic barriers that may limit educational options. Analyzing how factors

like access to quality education and socioeconomic disparities impact agency prompts a deeper examination of the structural inequalities that can lead to early educational disengagement. In this context, understanding agency as situated within historical and social contexts underscores the significance of addressing systemic inequities.

The principle of Time and Place aligns with the life course paradigm by underscoring the influence of historical context and location on individual trajectories (Hareven 1994; Bernardi et al. 2019). Applying this principle to ELET research involves investigating how past policies, economic fluctuations, and societal norms have contributed to early leaving rates. This perspective prompts consideration of the broader societal changes that may have shaped the educational choices of specific cohorts and calls for policies that address historical disparities.

The Timing principle, which acknowledges the interaction between individual experiences, family cycles, and historical events, offers a lens through which to analyze the timing of ELET (Elder et al. 2003). Economic shifts, family circumstances, and societal changes can converge to impact decisions regarding education. By scrutinizing how these factors interplay and their disproportionate impact on vulnerable populations, researchers gain insights into the complex tapestry of reasons behind ELET.

Linked Lives, the final principle, reinforces the interdependence of life courses, emphasizing how individual trajectories are shaped by the experiences of others (Elder et al. 2003). In the context of ELET, this principle invites examination of the role of communities, families, and social networks in shaping educational decisions. Investigating the influence of mentorship, role models, and support networks can shed light on potential strategies to mitigate the risk of ELET, particularly for marginalized individuals who may lack access to such resources.

In this way, the life course paradigm provides a comprehensive framework for understanding the intricate web of factors contributing to ELET. By utilizing this lens, researchers can move beyond surface-level analysis and delve into the underlying systemic issues that drive educational disengagement. As the paradigm encourages a holistic exploration of policy interventions, structural reforms, and community engagement strategies to address ELET and enhance educational equity for all individuals, irrespective of their background or circumstances.

## 6. Conclusions

In conclusion, viewing Early Leaving from Education and Training through the lens of the life course perspective yields a richer and more comprehensive understanding of this complex process. While various explanatory models have provided valuable insights, the life course perspective's emphasis on holistic trajectories, interconnectedness, agency, and the interplay of historical context and societal structures offers a nuanced approach that resonates with the goal of achieving equitable educational opportunities.

This paradigm's application to ELET research provides a holistic grasp of the multifaceted factors contributing to ELET, encompassing not only individual choices but also systemic barriers, historical influences, and community dynamics. By incorporating principles such as Lifespan Development, Agency, Time and Place, Timing, and Linked Lives, researchers can uncover the intricate interplay of individual experiences and societal forces that shape educational trajectories.

In a world marked by evolving socioeconomic dynamics, embracing the life course perspective provides a means to address the underlying inequalities that perpetuate ELET. This approach calls for policy interventions that not only acknowledge individual agency but also recognize the broader historical and structural contexts that influence educational choices. By adopting this comprehensive framework, educators, policymakers, and researchers can work collaboratively to develop strategies that promote inclusivity, reduce disparities, and ensure that no individual is left behind in the pursuit of education. Ultimately, the life course perspective stands as an essential tool for fostering a more just and equitable educational landscape for all.

**Author Contributions:** Conceptualization, L.G.-P. and M.T.S.; methodology, L.G.-P.; software, L.G.-P.; validation, L.G.-P. and M.T.S.; formal analysis, L.G.-P.; investigation, L.G.-P. and M.T.S.; resources, L.G.-P. and M.T.S.; data curation, L.G.-P.; writing—original draft preparation, L.G.-P.; writing—review and editing, L.G.-P. and M.T.S.; visualization, L.G.-P.; supervision, L.G.-P. and M.T.S.; project administration, L.G.-P. and M.T.S.; funding acquisition, L.G.-P. and M.T.S. All authors have read and agreed to the published version of the manuscript.

**Funding:** This research received no external funding.

**Institutional Review Board Statement:** Not applicable.

**Informed Consent Statement:** Not applicable.

**Data Availability Statement:** Not applicable.

**Conflicts of Interest:** The authors declare no conflict of interest.

## Appendix A

**Table A1.** Summary of content in articles included in the review.

| Reference | Summary |
|---|---|
| Alexander et al. (2001) | The study then investigates the risk factors that contribute to dropout, as well as the resources that support children's schooling at four distinct schooling benchmarks: first grade, the rest of the elementary school (years 2–5), middle school (years 6–8), and the ninth year (first year of high school for those promoted each year). It is noted that academic, parental, and personal resources play a role in shaping dropout prospects at each of these time points. Notably, resources measured early in a child's schooling, such as academic resources, have predictive value for dropout outcomes, much like resources measured later in their schooling journey. Furthermore, the article presents evidence that these resources accumulate and interact, influencing dropout risk. This holds true even when considering the risk associated with socioeconomic status (SES). The patterns observed are interpreted within the context of a life course perspective on the dropout process. This perspective considers the various factors that contribute to a student's decision to disengage from school over an extended period. Overall, the article sheds light on the complex interplay of sociodemographic factors and academic performance, as well as parental and personal resources, and how they collectively impact the likelihood of high school dropout. The study underscores the importance of understanding these dynamics from a life course perspective to develop effective strategies for preventing dropout and fostering academic success. |
| Chen and Kaplan (2003) | This article discusses a study conducted from a life course perspective using longitudinal panel data collected across three developmental stages: early adolescence, young adulthood, and middle adulthood. The focus of the study is to examine how early school failure influences the status attainment at midlife. The findings of the study suggest that the impact of early school failure on status attainment at midlife is not solely mediated by the number of years of education completed in early adulthood. Instead, the study identifies additional mediating pathways, specifically lower levels of mental health and higher rates of deviant behaviors in early adulthood, which also play a role in influencing status attainment. The study also points out that there is a modest residual direct effect of school failure in adolescence on status attainment at midlife. This effect can be explained by considering inherited or acquired cognitive abilities and motivational dispositions exhibited in early adolescence, which may have lasting implications for an individual's status attainment later in life. Overall, the study highlights the complex interplay of factors over an individual's life course, including early school failure, mental health, deviant behaviors, and cognitive abilities, in shaping their status attainment at midlife. |

**Table A1.** *Cont.*

| Reference | Summary |
|---|---|
| Entwisle et al. (2004) | The article delves into the motivations, employment patterns, and ultimate outcomes of individuals who leave high school without graduating but later earn GEDs or return to school to obtain diplomas. It underscores the importance of considering these factors and provides insights into potential strategies to support at-risk students in achieving high school certification.<br><br>The findings of the study indicate that students in Baltimore who eventually earn high school diplomas share similar demographic characteristics and school performance with students who complete high school in national studies. Additionally, the study reveals that temporary dropouts (those who returned to school) had more positive motivational qualities and were more frequently employed before they dropped out compared to the permanent dropouts.<br><br>The article concludes by discussing the implications of these findings for education policies. It emphasizes the significant role of employment and alternative educational pathways to high school certification for disadvantaged adolescents. The study suggests that understanding and addressing students' motivational characteristics and employment opportunities could potentially contribute to reducing the high school dropout rate and improving the educational outcomes of disadvantaged youth. |
| Entwisle et al. (2005) | This study investigates the impact of employment on the likelihood of high school dropout among students in Baltimore, a city with a high poverty rate and a significant dropout problem. The researchers examined the relationship between employment types and dropout risk among 15-year-olds and 16-year-olds. For 15-year-olds, those who held jobs typical of teenagers, such as lawn mowing or babysitting, had a significantly lower likelihood of dropping out compared to those who took on adult-type jobs in manufacturing or business. However, this trend reversed when students turned 16. At age 16, students with adult-type jobs were less likely to drop out than those with teen jobs.<br><br>The study also found that patterns of work beyond ages 15 and 16 influenced dropout risk. Specifically, students who had been retained in their grade but made a smooth and orderly transition into the workforce were less likely to drop out. In contrast, students who had been retained and experienced a disorderly transition into employment had higher dropout rates. These findings suggest that the impact of employment on dropout risk varies depending on the age of the students and the type of jobs they hold. Teen jobs seem to have a protective effect against dropout at age 15, while adult-type jobs become more beneficial in reducing dropout risk at age 16. Furthermore, a well-managed transition into work appears to be associated with lower dropout rates among retained students. The study highlights the importance of considering age and work patterns when examining the relationship between employment and dropout risk in high-poverty settings. |
| Pallas (2006) | The article addresses the need for a fundamental shift in how social scientists approach the study of the transition to adulthood. Traditionally, researchers have placed significant emphasis on school-leaving as a pivotal aspect of the transition to adulthood. However, the author contends that this perspective has its limitations. The article draws upon 51 retrospective life-history interviews conducted with adults from mid-Michigan and introduces an alternative viewpoint.<br><br>The author suggests that the conventional understanding of the transition to adulthood, which revolves around a series of role changes such as transitioning from a student to a non-student, is essentially a construct shaped by the viewpoints of social scientists. Nonetheless, this viewpoint falls short in adequately considering the perspectives of individuals who have personally undergone these transitions. Furthermore, the author highlights that the timing of individuals completing their education may not necessarily align with their perception of becoming adults. This challenges the conventional notion that adulthood is closely tied to milestones such as leaving school, entering the workforce, or starting a family.<br><br>Moreover, the article underscores a significant knowledge gap when it comes to individuals returning to education later in life. Framing adulthood primarily in the context of school departure fails to encompass this particular process. In conclusion, the author advocates for a more holistic approach to studying the life course, one that extends beyond mere acknowledgment of role transitions and instead integrates theories that take into account the subjective meanings and perspectives of individuals as they undergo these transitions. |

**Table A1.** *Cont.*

| Reference | Summary |
| --- | --- |
| Bersani and Chappie (2007) | This research highlights the importance of understanding turning points during adolescence and specifically focuses on school failure as a potential crucial event that can shape an individual's life trajectory, particularly in terms of involvement in delinquent activities. The findings of the research provide support for the notion that school failure indeed operates as an influential turning point during adolescence. Additionally, the study confirms that school failure is significantly influenced by factors from various domains, including structural, relational, and individual factors. While school failure might be commonly regarded as the outcome of a prolonged process of disengagement from academics, the research suggests that it plays a pivotal role as a negative turning point in an individual's life. |
| Janosz et al. (2008) | This article addresses the limited research on the prospective relationship between school engagement and dropout, which often overlooks the diverse academic and social backgrounds of students who leave school prematurely. To fill this gap, the study examines the various patterns of school engagement over time and their predictive associations with dropout. Using an accelerated longitudinal design, the researchers identified seven distinct trajectories of school engagement among 12- to 16-year-old students (N = 13,300). The majority of students fell into three stable trajectories, representing developmentally normative pathways characterized by consistently moderate to very high levels of school engagement. However, a smaller group of participants (about one-tenth) followed four nonnormative or unexpected pathways, which were associated with a significant proportion of dropouts. The risk of dropout was notably linked to trajectories of school engagement that exhibited significant fluctuations or instability.<br><br>In conclusion, the article discusses the importance of striking a balance between universal strategies and more targeted, differentiated approaches to prevent dropout. Additionally, the researchers highlight the need to explore why some students with high levels of school engagement still end up dropping out, even within normative trajectories. Understanding these complexities can inform more effective strategies to address the issue of dropout and enhance educational outcomes for all students. |
| Rumberger and Rotermund (2012) | This article reviews prominent models of school dropout, emphasizing the role of individual and contextual factors, particularly school engagement-related factors, in the dropout process. The models differ in terms of the specific factors influencing dropout and the underlying process leading to this outcome. The research literature indicates that multiple factors contribute to a student's decision to drop out, and it is not solely determined by school-related experiences but also by activities and behaviors outside of school, such as engaging in deviant and criminal behaviors. The article introduces the concept of life course models, which refer to long-term longitudinal studies in the USA that track the educational experiences and outcomes of children over time. One study conducted in Chicago found that family background, parental involvement, and the child's cognitive and behavioral performance in school predict dropout. Another study in California highlighted the influence of nonconventional family lifestyles and cumulative family stresses on dropout. The Beginning Baltimore Study proposed a life course perspective, viewing dropout as a long-term process of progressive academic disengagement, identifying factors such as students' school experiences, personal resources, and parental support that influence dropout across various developmental periods. Similarly, the Chicago Longitudinal Study revealed how preschool participation affects long-term outcomes, including educational attainment and juvenile delinquency. This impact is mediated through factors such as school performance, social adjustment, family support, school quality, and motivation. Overall, the article highlights the complexity and multifaceted nature of the dropout process, emphasizing the significance of school engagement-related factors and considering various life course models to better understand the factors influencing dropout and inform prevention strategies. |

**Table A1.** *Cont.*

| Reference | Summary |
|---|---|
| Schmid and Stalder (2012) | The article discusses the process of dropping out of vocational education and training (VET) programs in Germany and Switzerland. It highlights the shortage of training places in these countries, leading to intense competition among candidates. In Switzerland, approximately one in three VET applicants has to wait for at least a year to secure an apprenticeship placement after leaving school. The article also mentions the high dropout rate in apprenticeships, which poses challenges to educational policies. The article suggests that dropping out of vocational education and training (VET) programs should not always be viewed negatively, as it can provide an opportunity for individuals to address issues in their educational journey, improve their educational situations, and make realignments. Many of those who dropped out eventually switched to different companies, educational levels, or fields, and approximately two-thirds of them successfully completed their vocational education after making these changes. However, it's important to note that dropping out of vocational education and training (VET) programs is not a guaranteed positive outcome for everyone. For around one-third of the youths who drop out, it marks the end of their education at the upper secondary level in VET programs. The main focus of the chapter is to track the life paths of these dropouts, describe their educational situations during the first three years after leaving their apprenticeships, and discuss how dropping out can represent an opportunity for positive change in some cases. |
| Mínguez (2013) | This research investigates early school leaving (ESL) in five European countries (Germany, Denmark, Finland, Belgium, Spain, and the United Kingdom) using a comparative sociological approach. ESL is defined as the percentage of individuals aged 18-24 with at lower secondary education and not in further education or training. The study aims to analyse the main causes of ESL in these countries, considering factors like investment in education, gender, ethnic background, and family background. It also explores the employment status and training of young school leavers. The analysis is based on aggregate data from the European Labour Force Survey and OECD Data Base, with reference to the concept of "transitional life course regime" developed by Walther (2006) for each selected country. The findings reveal that countries with lower social spending on education, such as Spain, exhibit higher rates of school dropouts and youth unemployment compared to countries with stronger welfare states, like Finland and Germany. Differences in early school leaving rates are also evident between immigrants and nationals in all countries except the United Kingdom and Germany, with Spain experiencing more prominent disparities. The study highlights the significance of socioeconomic conditions and family background in influencing school leavers. The conclusion suggests that variations in ESL rates across European countries are influenced by social spending on education and the education policies implemented by states. The impact of gender and nationality also differs depending on the education model in place. The results indicate the need for European welfare states to strengthen their education policies, increase investment in education, and redesign their education systems to address and reduce early school leaving. |
| Pharris-Ciurej et al. (2012) | The authors highlight the discrepancy between reported high school graduation rates in the USA surveys and the actual rates recorded in administrative records. The study's use of individual-level administrative data provides a more accurate picture of high school attrition and reveals that factors such as the ninth grade shock play a significant role in students not graduating high school on time. This has implications for understanding the challenges students face in completing their high school education. |
| Ríos (2014) | This study examines the impact of inequalities related to social risks of origin and life course on academic events termed "educational risks." The analysis focuses on a cohort of Uruguayan students evaluated in 2006 by the Program for International Student Assessment (PISA) during the 10th grade. Using longitudinal methodology, including descriptive survival analysis and a discrete time logistic regression model, the study finds that educational trajectories involving "risk events" have significant implications for educational policy. Such events weaken pedagogical and social ties, leading to non-standardized educational paths. The study highlights the influence of normative age milestones for adulthood transitions on individual educational trajectories. Vulnerable groups, such as students from manual worker households or those with lower competencies, exhibit higher intensity and earlier occurrence of educational risk events. Addressing these risks requires policies integrating labor and educational trajectories and implementing a national care system to reconcile educational pursuits with parenting responsibilities. |

**Table A1.** *Cont.*

| Reference | Summary |
| --- | --- |
| Gibbs and Heaton (2014) | Preventing school dropout has been recognized as a crucial aspect of achieving the Millennium Development Goals. Despite achieving universal primary school enrolment, Mexico still faces a high dropout rate of approximately 50% by the end of formal schooling. To investigate this issue, a unique and nationally representative dataset, the Mexico Family Life Survey, was utilized to track children aged 5 to 11 in 2002 until 2005-2006 to identify dropout occurrences. Applying a life course perspective, the study examined the interactions between family, school, and macro-factors concerning the child's schooling level and the transition from primary to secondary school. Results revealed that the transition to secondary school showed the highest dropout rates, with rurality significantly affecting this phase. Family factors emerged as the most predictive indicators of dropout, with the roles of maternal education fading over time, while the influence of an unemployed father grew. |
| Dupéré et al. (2015) | This research explores the concept of high school dropout and challenges the traditional perspective that views dropout because of long-term academic struggles and disengagement. Instead, the article highlights the heterogeneity of pathways leading to dropout, suggesting that some students may leave school in response to specific late-emerging situations, while others persist despite earlier difficulties. The authors propose a stress process, life course model of dropout, which aims to integrate both long-term vulnerabilities and immediate disruptive events and contingencies that contribute to students leaving school. This model acknowledges that dropout is not solely influenced by prolonged academic difficulties but also by critical events and circumstances occurring during a student's high school career. Moreover, the study emphasizes the importance of considering socioeconomic conditions, geographical factors, and historical contexts when examining the determinants of dropout. These external factors can shape the dropout experience differently for students in various contexts. The adoption of the stress process, life course model allows researchers and policymakers to gain a more comprehensive understanding of the complexities surrounding high school dropout. By considering both long-term and immediate determinants, this model can inform the development of targeted interventions and policies aimed at reducing dropout rates and promoting educational attainment for all students. |
| Campbell (2015) | This article delves into sociological research on high school dropouts, focusing on two key aspects: the reasons for dropping out and the resulting consequences. While prior studies have primarily centred on identifying dropout determinants and examining the demographics of dropouts, limited attention has been given to understanding the outcomes experienced by dropouts after leaving school. The article discusses potential factors contributing to the negative consequences faced by dropouts, including variations in human capital, signalling theory, and social closure. Furthermore, the article acknowledges the empirical challenges inherent in studying the effects of dropping out, mainly due to inherent differences between dropouts and high school graduates. Looking ahead, the article emphasizes the importance of future research in unravelling the mechanisms connecting dropping out to unfavourable long-term outcomes. Additionally, it underscores the significance of investigating the heterogeneity of dropout effects, considering factors such as local labour market conditions and individual-level characteristics. The dynamic nature of the labour market and the evolving population of dropouts raise pertinent questions about the changing consequences of dropping out over time and the influence of reasons for leaving school on later life outcomes. Furthermore, the article highlights the need to broaden the examination of disadvantages faced by dropouts beyond mere labour market penalties. It suggests exploring the impact of assortative marriage, fertility rates, health disparities, and limited financial strategies on the multifaceted disadvantages experienced by high school dropouts. A theoretical understanding of the intricate connections between dropping out and multiple adversities is deemed necessary to comprehensively analyse the consequences of this educational outcome. |

**Table A1.** *Cont.*

| Reference | Summary |
| --- | --- |
| Vogt (2017) | The author explores the issue of early school leaving in upper secondary education through the lens of life course theory, focusing on age norms. It investigates how early school leaving is defined concerning chronological age and how it reflects specific norms regarding the timing of life events. The study highlights that as early school leaving gained prominence on the international policy agenda in the 2000s, young people's trajectories started to be compared against the norm of prolonged and orderly transitions from school to work, predominantly in the academic track. Transition patterns common within vocational education, leading to qualifications obtained later in life, may be considered problematic under these age norms. Pupils in vocational tracks often follow less standardized routes not strictly aligned with age norms. Consequently, early school leaving can be perceived as a form of deviance, resulting from the universalization of age norms that favour academic pathways. The article suggests that age norms play a significant yet often overlooked role in shaping school-to-work transitions in contemporary contexts. Understanding the impact of age norms on early school leaving is crucial to developing more comprehensive strategies to address this issue. |
| Holtmann et al. (2017) | The article addresses the challenges that young individuals with low educational attainment face when transitioning from school to the workforce. These individuals often encounter difficulties in finding suitable employment due to their limited educational background. The authors emphasize that while this group of school leavers with low achievement appears homogenous, they exhibit variations in terms of personal and social resources. These differences may impact their ability to take initiative during the transition period and consequently influence how potential employers perceive their potential contributions in the labor market. The study aims to investigate the interaction between personal agency (the ability to take actions and make choices) and structural constraints in the context of low-achieving adolescents' transitions from school to work. The authors explore whether adolescents with higher cognitive and noncognitive skills, as well as greater parental resources, have improved opportunities during this transition phase. Additionally, they examine whether the persistent disadvantages faced by low-achieving school leavers are a result of their low educational attainment or if other factors come into play. The research findings indicate that the transition period offers new possibilities primarily for those low-achieving adolescents who have better vocational orientation and higher career aspirations. These individuals tend to put in greater efforts during the application process, leading to better outcomes. Interestingly, the success of the initiative taken by these adolescents is found to be influenced by the type of school-leaving certificate and the school attended, but not significantly affected by competences, noncognitive characteristics, or parental background. This suggests that the label of having low qualifications itself acts as a significant obstacle during the transition period, particularly for the subgroup with the lowest educational attainment. These individuals struggle to access the necessary training for economic independence and face challenges in their overall transition to adulthood. |

*Source.* Own elaboration.

**Appendix B**

**Table A2.** Analysis of how life course is approached in the articles included in the review.

| Reference | Summary |
| --- | --- |
| Ríos (2014) | The article explores the concept of risks within the framework of the life course paradigm. In essence, Ríos (2014) links the concept of risks to the life course paradigm by highlighting how risks intersect with pivotal life stages, shaping individual life trajectories and opportunities. The article's focus on dynamic interactions aligns with the principles of the life course paradigm, illustrating how events, states, and social contexts collectively influence individual experiences. |
| Gibbs and Heaton (2014) | Gibbs and Heaton (2014) utilizes longitudinal data to analyze the impact of educational transitions on dropout from a life course perspective. This perspective emphasizes the dynamic interplay of individual characteristics, family, school, and broader social changes that shape trajectories of experience. By considering transitions as a process rather than isolated events, the study captures the shifting ecological context influenced by family, community, and school throughout a child's education journey. |

| Reference | Summary |
|---|---|
| Dupéré et al. (2015) | Dupéré et al. (2015) integrates the concepts of the stress process and life course perspectives within the context of high school dropout and the transition from adolescence to early adulthood. The article, that is an extensive review of models, acknowledges the complementarity of these two models and their potential to provide a comprehensive understanding of dropout behavior. The stress process model, primarily applied to study mental health issues, is expanded to encompass high school dropout. This model conceptualizes stressors as problematic external circumstances that challenge individuals' adaptive capabilities and contribute to adjustment problems. Stressors can be disruptive events or prolonged difficulties. The stress process framework emphasizes the indirect relationship between stressors and individual adjustment, often involving secondary stressors that stem from the primary stressor. Stress proliferation, the tendency for stressors to cluster and create cumulative adversity, is highlighted. This perspective encourages a consideration of the entire configuration of stressors an individual experiences over time. The life course model, on the other hand, examines the unfolding of lives within specific historical and geographical contexts. This model captures both long-term antecedents of important life transitions and their enduring consequences. The life course approach considers the broader developmental trajectory beyond just health outcomes and emphasizes the interplay between historical, geographical, and developmental factors. Dupéré et al. propose an integrative model that applies both the stress process and life course perspectives to the study of high school dropout. They highlight the relevance of the stress process model in the education context by drawing parallels between dropout and stress-related processes. Dropout is viewed as a retreat in the face of stressful or defeating social situations, analogous to depression or other mental health issues. Stressors are identified as significant contributors to dropout behavior, and the concept of stress proliferation underscores how multiple stressors can accumulate and impact educational outcomes. The article emphasizes that considering the full configuration of stressors is crucial for understanding the complexities of dropout. Furthermore, the article suggests that the unequal distribution and configuration of stressors across socioeconomic lines can shed light on the mechanisms linking poverty and high school dropout. Disadvantaged individuals may be exposed to a greater variety of stressors, potentially affecting their educational attainment. By integrating both models, the article aims to provide a comprehensive framework for studying high school dropout that considers the interplay between individual stressors, developmental trajectories, historical context, and socioeconomic factors. |
| Janosz et al. (2008) | The article by Janosz et al. (2008) aligns with the life course paradigm by investigating developmental trajectories of school engagement and their associations with school dropout. The authors acknowledge that traditional approaches often compare average characteristics of high school completion with non-completion or focus on general relations between predictors and dropout. However, they emphasize the importance of considering specific developmental trajectories grounded in established risk factors, which aligns with the life course paradigm's emphasis on individual trajectories shaped by various influences. The study identifies both normative and non-normative trajectories of school engagement. They find a normative trajectory characterized by high and stable school engagement, shared by half of the sample. This trajectory is consistent with the life course paradigm's principle of lifespan development, recognizing cumulative life experiences. The authors also consider the variations in stability, connecting them to individual and contextual factors, reflecting the principle of agency within the life course paradigm. The authors address the complex nature of stability in engagement, suggesting it might reflect both individual potentials and underlying support from the family or school environment. This resonates with the life course paradigm's emphasis on interconnectedness between individual lives and social contexts. Furthermore, they highlight sex differences in engagement trajectories and dropout risk, with boys more likely to follow unstable trajectories. This aligns with the life course paradigm's principle of time and place, acknowledging the influence of historical and social context. The authors suggest that the dynamic processes underlying various trajectories of school engagement need further investigation, like how the life course paradigm focuses on understanding the interactions between individual lives and broader societal changes. In conclusion, the study's exploration of developmental trajectories of school engagement, their associations with dropout, and the consideration of individual and contextual factors align well with the principles and perspectives of the life course paradigm. The study's findings contribute to the understanding of the complexities of school engagement and dropout within a holistic life course perspective. |

**Table A2.** *Cont.*

| Reference | Summary |
| --- | --- |
| Rumberger and Rotermund (2012) | In the article by Rumberger and Rotermund (2012), they discuss the application of life course models in understanding the dropout process in education. These models are derived from long-term longitudinal studies conducted in the USA that track the educational experiences and outcomes of children. The term, "life course models", is used because of the studies' long-term perspective. Although the article doesn't entirely focus on the life course paradigm, it presents examples of how researchers have employed longitudinal studies and life course models to understand the complex factors influencing the dropout process in education, taking into account various developmental periods and influencing factors. |
| Entwisle et al. (2005) | In the article, the authors use the life course paradigm to explore the relationship between work patterns and school dropout rates among disadvantaged urban students. The article investigates how the sequence of jobs over ages 15 and 16 affects dropout rates and students' ability to balance work and school demands.<br>Drawing from life course ideas, the authors hypothesize that students who transition into work in an orderly manner should have an easier time managing the demands of both work and school compared to those with disorderly transitions. For instance, if a teenage job at age 15 precedes an adult job at age 16, it reflects a more orderly pattern that is easier to manage, given that teenage jobs generally offer more flexibility and control over working hours. On the other hand, disorderly patterns like adult work at age 15 followed by teenage work at age 16 are seen as contrary to developmental expectations and potentially disruptive. |
| Campbell (2015) | Campbell (2015) engages with the life course paradigm by applying a perspective that goes beyond a narrow focus on push and pull factors for understanding school dropout. The authors highlight that students who leave school are exposed to various risks that contribute to their decision to drop out. They introduce the life course perspective to illustrate how events and processes starting from early life experiences and continuing throughout one's academic career can culminate in dropping out. |
| Chen and Kaplan (2003) | In the study conducted by Chen and Kaplan (2003), the researchers employed a life course perspective to investigate the impact of early school failure on status attainment in midlife. The study utilized longitudinal panel data collected at three developmental stages: early adolescence, young adulthood, and middle adulthood. Through structural equation analyses, the researchers aimed to uncover the ways in which early school failure influences an individual's status attainment later in life. |
| Vogt (2017) | Vogt (2017) investigates the implications of the concept of Early School Leaving (ESL) for norms and expectations related to life course transitions from school to work. Using life course theory on age norms, the authors argue that the contemporary framing of transition patterns in terms of ESL implicitly establishes norms for the timing of events across an individual's life course. By designating appropriate behaviors within certain age ranges and what is not appropriate, the concept of ESL has likely contributed to narrowing the age norms associated with young people's school to work transitions. |
| Mínguez (2013) | The study employs a comparative multi-theoretical viewpoint, drawing on the concept of "transitional life course regime" developed by Walther (2006). This concept categorizes countries based on their welfare systems, educational policies, and job markets, which impact young people's transitions. The article uses aggregate data from sources like the European Labour Force Survey and OECD Data Base.<br>The research highlights the importance of socioeconomic conditions and family background on early school leavers, emphasizing that educational inequality based on nationality and family origin is more pronounced in countries with weaker welfare state involvement in public education. The study underscores the need for European welfare states to strengthen their education policies and redesign their education systems to address early school leaving.<br>Overall, the article demonstrates how the life course paradigm is applied to understand the influences on early school leaving across different European countries. It contextualizes individual behavior within broader national and institutional frameworks and highlights the significance of socioeconomic factors in shaping educational outcomes. |

**Table A2.** *Cont.*

| Reference | Summary |
|---|---|
| Entwisle et al. (2004) | The article discusses how several existing studies have explored predictors of high school dropouts obtaining GEDs or returning to school for diplomas. These predictors include socioeconomic status, academic performance, parenthood status, and students' expectations. However, the article highlights that these studies have not fully considered the role of students' motivational characteristics and employment patterns before dropping out.<br>To address this gap, the article takes a "life course perspective" and draws on a longitudinal study conducted in Baltimore, a city known for its high dropout rate (over 40%). The study compares two groups of dropouts: those who dropped out temporarily and those who dropped out permanently. The goal is to understand the differences between these groups and identify factors that might influence their later success in achieving high school certification. |
| Schmid and Stalder (2012) | The article's analysis is rooted in the life course paradigm, as it focuses on the trajectories of young individuals who drop out of apprenticeship training. It explores the experiences of these "education returners" over a three-year period after dropping out. The study finds that three-quarters of the dropouts continue their education within three years of leaving the apprenticeship, indicating that dropping out of education is not necessarily negative and can lead to opportunities for change and improvement in educational situations.<br>The life course perspective is evident in the article's examination of the educational paths that these dropouts take after leaving apprenticeship training. It delves into how dropping out can lead to reentry into education, as well as shifts to different companies, educational levels, or fields. |
| Pharris-Ciurej et al. (2012) | The author engages with the life course paradigm to understand the process of high school dropout. The life course perspective is introduced as a conceptual breakthrough that seeks to analyze events and influences across the span of years from childhood to adolescence. The central argument of this perspective is that dropping out of school is not a single isolated event; instead, it is the result of a broader process of academic failure and disengagement that starts early in a student's academic journey.<br>The article emphasizes the interconnectedness of various factors that contribute to academic failure and dropping out, starting from distal forces like family background and pre-school experiences, which can shape and condition the more immediate events leading to dropout. Social background's impact on educational continuation changes at different stages of the educational journey. The influence of factors like socioeconomic status, educational resources, parental attitudes, and family structure are explored through the lens of the life course paradigm.The life course framework is used to establish the temporal order of family background variables and early life experiences, often gathered from retrospective survey questions. However, the challenge of recall in retrospective surveys is acknowledged, particularly when it comes to timing and specifics of prior academic performance and attendance. To mitigate this issue, the author highlights the importance of prospective studies or longitudinal data collection to capture contemporaneous reports of school experiences.<br>Methodologically, the article draws upon the life course perspective and previous research to model the determination of on-time high school graduation for ninth graders. Logistic regression equations are employed, considering a range of independent variables to understand their associations with high school graduation. The analysis acknowledges that high school graduation may encompass both in-district graduation and dropout but argues that the associations between independent variables and the outcome should still be valid due to the similarities between dropping out and transferring to another district. |

**Table A2.** *Cont.*

| Reference | Summary |
| --- | --- |
| Bersani and Chappie (2007) | Bersani and Chappie (2007) engage with the life course paradigm by exploring the concept of turning points in individual trajectories and their impact on life course trajectories. The article emphasizes that life course theory focuses on specific life transitions or turning points that can significantly alter an individual's trajectory, invoking substantial change. The theoretical framework of the life course paradigm is built on the concepts of trajectories (long-term patterns of behavior) and transitions (specific life events occurring over shorter periods of time).<br><br>Moreover, this paper highlights the importance of turning points, which are catalysts for change in life course trajectories. It notes that positive turning points, such as marriage, employment, parenthood, and others, can shift an individual's trajectory from deviant to normative paths. However, the article also acknowledges that turning points are not limited to positive changes; negative events during adolescence, like school failure, can also be turning points that potentially lead individuals towards delinquency.<br><br>The article also argues that adolescence, due to its dynamic nature and transition into young adulthood, presents a significant opportunity for the occurrence of turning points. It suggests that negative events during adolescence, like school failure, can set individuals on deviant trajectories, and this aligns with control theories that emphasize weakened social bonds leading to delinquency during childhood and adolescence. It suggests that school failure can result in being "off time" in maturational development, leading to adverse consequences in adulthood. It also notes that remaining on deviant trajectories due to school failure can limit individuals' opportunities for a conventional life and increase involvement in deviant and criminal behavior. |
| Pallas (2006) | In Pallas' (2006) article, "The Construction of Roles in the Life Course," the author employs the life course paradigm to study the effects of schooling on individuals over time. Pallas emphasizes the multidisciplinary nature of the life course approach, focusing on individual development from birth to death. The key features of the life course perspective include:<br><br>1. Emphasis on Time: Pallas highlights the significance of time as a means of organizing the life course. This encompasses biological, social, and historical time. Biological time relates to the pace of human development, while social time refers to age-appropriate behavior based on societal norms. Historical time includes historical events that shape an individual's development within the broader social, political, and economic contexts.<br>2. Continuity and Change: Pallas discusses the concept of continuity and change across different life phases such as infancy, childhood, adolescence, adulthood, and old age. The perspective examines how experiences in one phase of life influence subsequent phases and explores the idea of critical periods in development.<br>3. Individual Agency and Social Forces: Pallas presents individuals as both producers of their own development and subjects influenced by social forces. While social contexts like family, school, and workplace are significant agents of socialization, Pallas suggests that much of the existing work has focused on these contexts.<br>4. Roles in Context: The article discusses the construction of roles in the life course, emphasizing how social roles individuals assume as they age contribute to their identity and are often associated with specific social institutions.<br><br>The application of the life course paradigm to education is evident in the discussion of roles and stages in the life course. Pallas explores how individuals transition from one role to another as they age, noting that school leaving can be a crucial marker in this transition, particularly from adolescence to adulthood. Pallas argues that leaving school signifies the transition from a role of dependence to one of independence, as it indicates financial and socioemotional independence from the family of origin.<br><br>However, the article also acknowledges that the changing configuration of roles in society challenges the traditional view of school-leaving as a key indicator of independence and the onset of adulthood. With more individuals combining roles such as student, worker, spouse, and parent, the article questions the utility of school-leaving as a definitive marker. The increasing prevalence of "nontraditional" students beyond the traditional 18–24 age range in higher education raises further questions about the role of education in the life course. |

**Table A2.** *Cont.*

| Reference | Summary |
| --- | --- |
| Alexander et al. (2001) | The article by Alexander et al. (2001) examines the dropout process in the context of the life course paradigm. The authors emphasize that dropout is not merely an "event" but a "process" of progressive academic disengagement that often traces back to children's earliest experiences in school. They argue that dropout can be better understood by considering the various spheres of influence, such as school, home, and community, that shape children's academic development. The study observes that dropout is influenced by multiple factors that develop over time, often starting in the early grades of school. Children who exhibit weak school performance, grade retention, aggressive behavior, and experience stressful family changes in the early grades are more likely to be at risk for later dropout. These early factors can set the trajectory for dropout by influencing students' attitudes, behaviors, and sense of academic self-identity. |
| Holtmann et al. (2017) | This article contributes to understanding how the life course paradigm applies to the specific context of school-to-work transitions for individuals with low educational attainment, highlighting the role of agency, vocational orientation, and the impact of labeling. |

*Source.* Own Elaboration.

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
