# Peer review of "Early Leaving from Education and Training and Related Matters through the Lens of the Life Course Paradigm: A Systematic Review of the Literature"

_socsci, doi:10.3390/socsci12090521_

Round 1
Reviewer 1 Report
The topic of the article is relevant to the planned issue of Social Sciences on early school leaving. Revealing the heuristic potential of the life-course paradigm for understanding early school leaving (ESL) could provide valuable insights for better grasping its essence, determinants, and consequences. Empirically, the paper is based on a systematic review of 11 articles.
However, at this stage, the paper has some shortcomings which refer to the coherence between its aims and the suggested study design, its methodology, and the discussion of findings.
Below are some more concrete comments:
1) It remains unclear how ESL is defined. It seems that the notion of ESL is used interchangeably with the notions of dropout and school failure. If this is the case, this should be explained.
2) In section 2.1. Early school leaving and its portrayal in the literature are presented, drawing on three other articles, 8 models of explanation of dropout. However, they are not critically discussed and their strengths and weaknesses are not revealed against the life course paradigm, which is the focus of this paper.
3) It is argued that “[t]he life course paradigm presents significant potential for understanding early school leaving” (lines 291-292) but this main thesis should be better developed and substantiated. For example, how the five fundamental principles of the life course paradigm could be applied as a prism for a critical review of the existing literature on ESL and for a deeper understanding of ESL?
4) The empirical basis of the study consists of 11 articles that, according to the adopted criteria, investigate ESL from the life course paradigm as an explanatory model or approach. However, I am not convinced that the selection process is fully correct. On the one hand, it is not obvious that some of the articles included in the analysis, for example, Hirschfield, P. (2009) and Minguez (2013), use the life course paradigm in their theoretical framework. On the other hand, some other articles have not been included in the analysis, although they meet the selection criteria. I have done a quick check and below are some examples:
- Ensminger, M. and Slusarcick, A. (1992). Paths to high school graduation or dropout: A longitudinal study of a first-grade cohort, Sociology of Education, 65, 95-113.
- Ross, A. and Leathwood, C. (2013) ‘Problematising early school leaving’, European Journal of Education 48(3): 405–18.
- Alexander, Karl, Doris Entwisle, and Nader S. Kabbani. 2001. ‘The Dropout Process in Life Course Perspective. Teachers College Record 103: 760–82.
5) It is stated that “the analysis of bibliometric information reveals a significant academic interest in investigating early school leaving from a life course theoretical-methodological paradigm (lines 380-381). But how this conclusion is made as only 11 articles were found for the period of 1990-2018 which analyse ESL from a life-course perspective?
6) I think that all summaries (Table 2: Summary of content in articles included in the review) should be written in another way – they should show how the application of the life course perspective enhances the understanding of ESL
7) Some technical remarks:
- ESL as an abbreviation is introduced in line 337, while it has been used in the previous pages extensively,
- Vogt is cited as Vogt (2017) in Table 2 and as Vogt (2018) in the references and in other parts of the paper.
Editing by a native English speaker is needed.
Reviewer 2 Report
It is a very relevant paper. There are a few things you may want to improve. Find some suggestions below, which I think you may introduce without too much effort. The only real concern I have is about the conceptual framework:
1 - In the theoretical framework, you need to start by explaining the concept of ESL. It is very specific. Even if the study is about ESL, it might be interesting to introduce at least a footnote that recognizes the existance of the indicator Early Leaving from Education and Training (which tries to shelter the diversity of the education offer in our time). You report to may studies on dropout and drop-out. You must make sure what the authors are talking about. Both these concepts are distinct from ESL. You may need to swap the title to include the different ways young people leave school. Otherwise some studies do not seem to fit in the problem you wnat to dicsuss.
2 - When going through your paper and references I noticed you also make reference to studies on school failure and so forth. It might be relevant to address the nuances of these concepts to underline the focus of your study. As you know, ESL refers to a very specific age group, for example. You must avoid theoretical deviations.
3 - When you make correlations between ESL and the life course perspective you may lead readers into mistake because you are not just talking about ESL bu about a diversity of ways young pepole leave school. Must make sure what you are really talking about.
3 - You may want to clarify the value of education in itself and beyond its relation with the world of economy. The atractiveness of the educational offer together with positive teacher-student realtions, for example, have been identified in several studies (which you do not refer) as pull factors.
4 - When referring to 'factors' please remember to refer the worldwide 'economic/ competitive (macro-systemic) factor' that leaves many young people behind.
5 - You quite rigthly posed ESL as a process. So, make sure you not refer to it as a phenomenon (here and there in the article & conclusions)
6 - You need to complete the idea on line 267...
7 - Methodological section is really OK
8 - In the second paragraph of your conclusion you seem to claim that ESL must be reconceptualized. I think this has been done already by means of the introduction of ELET, as an indicator in the most recent studies and policies. May be you need to say it differently.
9 - Paragraph 3, seems still argumentative, may be you can change the style of writing because conclusions should be conclusive (arguments were developed throughout).
Round 2
Reviewer 1 Report
The present paper is a revised version of a paper which has been submitted earlier and reviewed. The revised version has addressed adequately most of the issues raised in the previous review.
My recommendation is for the paper to be published in its present version.
Author Response
Thank you very much for accepting our manuscript.
Best regards,
Authors
Reviewer 2 Report
Early leaving FROM education and training - is the correct expression. Please change throughout the article.
2 - Understanding Early Leaving from Education and Training Through the Lens of the Life Course Paradigm. If you change the title a bit it would be more faithful to what you have done. I suggest: Understanding Early Leaving from Education and Training AND RELATED MATTERS...
Good theoretical clarification. To support potential readers, it might be useful to give examples of expressions that have been used together with ELET or confused with it (because you explore dropout, drop-out, early school leaving, and so forth that is not exactly the same, as I argued earlier).
167 - Tedesco's explanatory model of school failure AND EARLY SCHOOL LEAVING
178 - Escudero's explanatory model of SCHOOL FAILURE AND desertion
308 - instead of "A position, often interconnected" THIS POSITION IS OFTEN... (easier to read).
326 - underexplored in ELET AND RELATED literature
402 - Table 1
Bersani & Chapple 2007 School Failure as an Adoles-cent Turning Point Empiric n.d. What do you mean by n.d. (is it multidisciplinary? It has to have some kind of theoretical roots - please check)
407 ELET AND ITS RELATED MATTERS within... (I am trying to give you a way out in what concerns the indiscriminate use of different terms that do not mean exactly the same - you need to sort this out yourself throughout the following sections. the use of the expression 'and other related matters' may be useful)
411 - ELET THIS SUBJECT
438 the analysis of the content CONTENT ANALYSIS
440 - ELET LEAVING SCHOOL
502 4.3. ELET AND ITS RELATED MATTERS through Life...
503-554 - Good!
609-758 - Good! Make sure you use the right terms (You are not always talking about ELET)
700 Having exploreD - The use of English in the sentence does not sound exactly right. Please verify.
760 - In conclusion,
Conclusions OK
Appendixes OK
Please review the use of English. There are some spelling and structural mistakes
It is Ok. Here and there there are some spelling mistakes; some of the sentences are a bit too long and would benefict from being divided into two. Particularly in the apeendixes the structure of the language is quite 'basic'. I mean, for example, the repetition of 'the article discusses' and similar situations.
Author Response
Dear Reviewer,
We sincerely appreciate your meticulous review of the manuscript, as it has significantly aided us in enhancing the structure, content, and English language usage. During this final review, we have diligently addressed every aspect you mentioned. The requested modifications for English expression have been highlighted in yellow.
Warm regards,
The Authors